# Pulse potential mediated selectivity for the electrocatalytic oxidation of glycerol to glyceric acid

Wei Chen [1,2,4], Liang Zhang[1,3,4], Leitao Xu[1,4], Yuanqing He[1,2], Huan Pang [2] ✉, Shuangyin Wang [1] & Yuqin Zou [1] ✉

Preventing the deactivation of noble metal-based catalysts due to self-oxidation and poisonous adsorption is a significant challenge in organic electro-oxidation. In this study, we employ a pulsed potential electrolysis strategy for the selective electrocatalytic oxidation of glycerol to glyceric acid over a Pt-based catalyst. In situ Fourier-transform infrared spectroscopy, quasi-in situ X-ray photoelectron spectroscopy, and finite element simulations reveal that the pulsed potential could tailor the catalyst's oxidation and surface micro-environment. This prevents the overaccumulation of poisoning inter-mediate species and frees up active sites for the re-adsorption of OH adsorbate and glycerol. The pulsed potential electrolysis strategy results in a higher glyceric acid selectivity (81.8%) than constant-potential electrocatalysis with $0.7\ V_{RHE}$ (37.8%). This work offers an efficient strategy to mitigate the deactivation of noble metal-based electrocatalysts.

To alleviate the energy crisis, renewable bioenergy is an ideal and effective alternative to fossil energy, and the use of renewable bioenergy is consistent with the concept of sustainable development and environmental friendliness. Biodiesel production leads to the generation of large amounts of glycerol (GLY, ~10 wt.%) as byproducts[1]. Because of the low utilization efficiency of GLY, its demand is much lower than its production, leading to a continuous decline in its price (~0.17 US$·kg$^{-1}$ in 2019)[2]. In contrast, most GLY derivatives, such as glyceraldehyde (GLAD), dihydroxyacetone, glyceric acid (GLA), lactic acid (LA), and glycolic acid (GA), have high economic value[3,4]. Therefore, using the excess and low-cost GLY to obtain high-value-added products is crucial. Compared with traditional oxidation methods, electrocatalytic oxidation offers numerous outstanding advantages, such as the nonrequirement of chemical oxidants, mild reaction conditions, and controllable activity and selectivity[5,6]. However, electrocatalysts, particularly those based on noble metals, are prone to deactivation during the conventional constant-potential electrocatalysis (CE) process, causing low selectivity and poor stability.

To address this issue, the catalyst design and optimization of the electrolysis system are crucial. Pulsed electrocatalysis (PE) offers a simpler and more effective solution without complex electrocatalyst preparation and pre-treatment. PE can prevent electrocatalyst deactivation by altering the reaction environment, controlling catalyst restructuring, redistributing surface species, and adjusting interfacial pH[7]. Thus, PE has attracted much attention in numerous fields involving the electrocatalytic conversion of small molecules, such as $CO_2RR$[7-10], NORR[11-13], ORR[14], and water electrolysis[15], and so on. However, the application of PE in the electrocatalytic conversion of organic molecules remains limited.

Noble metals (e.g., Pt, Pd, and Au) exhibit excellent glycerol electro-oxidation reaction (GEOR) activity, such as a low onset potential, high selectivity for C3 products, and good corrosion resistance[16-18]. Among these, platinum is the most popular catalyst for glycerol oxidation owing to its strong catalytic activity in both acidic and alkaline environments[17]. However, applying an oxidation potential and prolonged electrolysis can result in PtO$_x$ formation, hindering the

[1]State Key Laboratory of Chem/Bio-Sensing and Chemometrics, College of Chemistry and Chemical Engineering, Hunan University, Changsha 410000, P. R. China. [2]School of Chemistry and Chemical Engineering, Yangzhou University, Yangzhou 225009, P. R. China. [3]Key Laboratory of Leather of Zhejiang Province, Institute of New Materials and Industrial Technologies, Wenzhou University, Wenzhou, Zhejiang 325035, P. R. China. [4]These authors contributed equally: Wei Chen, Liang Zhang, Leitao Xu. ✉e-mail: panghuan@yzu.edu.cn; yuqin_zou@hnu.edu.cn

adsorption of OH adsorbate ($OH_{ad}$) and alcohol substrate[19,20]. Additionally, during the GEOR process, intermediate species can adsorb and accumulate on the Pt site without dissociation and desorption, leading to electrocatalyst poisoning[21,22]. This hampers the re-adsorption of $OH_{ad}$ and GLY on the catalyst, resulting in low current density and selectivity for C3 products. To prevent the deactivation of Pt-based catalysts, extensive work has been conducted, such as the tuning of the $d$-band of Pt through alloying with a second metal element[18,21,22], and Pt loading onto oxide substrates to accelerate the removal of adsorbed intermediate species[23–25]. Although these methods can mitigate the inactivation of Pt catalysts to some extent, they are complex and resource-intensive, limiting their large-scale application.

In this work, the selective electrocatalytic oxidation of GLY to GLA was achieved through PE, with Pt nanocrystals capsuled in graphitic carbon (Pt@G) as the catalyst. The short duration of GLY oxidation potential (0.7 $V_{RHE}$) prevented the accumulation of poisoning intermediate species and the over-oxidation of the Pt@G catalyst. The application of relatively low potentials (0.3 $V_{RHE}$) facilitated the rapid desorption of GLA, allowing it to diffuse into the solution. Preventing the catalyst from reaching an over-oxidized state and releasing active sites on the catalyst surface enables the re-adsorption of $OH_{ad}$ and GLY, which alleviates the poisoning of the catalyst and improves the selectivity of GLA (81.8%) compared with that achieved in conventional CE (37.8%). This work provides a strategy to mitigate the deactivation of noble metal catalysts and achieve highly selective electrocatalytic oxidation of GLY to C3 products.

## Results

### Limitations of CE for GEOR over a Pt-based catalyst

Platinum-based catalysts are well known for their excellent catalytic activity in alcohol oxidation. The cyclic voltammogram of Pt@G in a 1 M KOH solution (Fig. 1a, black curve) exhibits the characteristic pattern of polycrystalline Pt. According to the literature, the potential range of 0–0.2 $V_{RHE}$ corresponded to the adsorption/desorption of $H_{ad}$, while the peaks between 0.2 and 0.4 $V_{RHE}$ corresponded to anion or proton adsorption/desorption, and reversible $OH_{ad}$ formation occurred between 0.7 and 0.9 $V_{RHE}$[26,27]. Upon the addition of 20 mM GLY, the peak around 0.2 $V_{RHE}$ disappeared in the second cyclic voltammetry (CV) cycle (Supplementary Fig. 1), indicating that the absorbed intermediate products limited the adsorption/desorption of $H_{ad}$. A noticeable oxidation peak occurred at 0.7 $V_{RHE}$ (Fig. 1a, red curve) owing to glycerol oxidation. As the CV curve continued to sweep positively, a significant drop in current density occurred. The LSV curves with different rotation speeds and GLY concentrations illustrate that the decrease in current density did not originate from mass transfer limitations (Supplementary Fig. 2). When the CV curve was swept negatively, an oxidation peak appeared at 0.8 $V_{RHE}$, coinciding with the desorption of $OH_{ad}$. This indicates that $OH_{ad}$ desorption released active sites for glycerol adsorption, resulting in a noticeable increase in oxidation current starting from 0.7 $V_{RHE}$. These phenomena suggest that the intermediate products of glycerol oxidation and $OH_{ad}$ occupied the active sites of the catalyst without desorption, leading to catalyst deactivation.

This phenomenon can also be confirmed through operando electrochemical impedance spectroscopy analysis. According to the literature, GEOR occurs at the low-frequency interface in Bode plots for Pt-based catalysts[28]. As shown in Supplementary Fig. 3, the phase angle decreased as the potential increased from 0.3 to 0.7 $V_{RHE}$. Similarly, the Nyquist plots (Fig. 1b) exhibited decreasing semicircle diameters, indicating lower charge-transfer resistance ($R_{ct}$) (Fig. 1c) and faster GEOR reaction kinetics as the potential increased from 0.3 to 0.7 $V_{RHE}$. However, with a further increase in potential (0.8–1.0 $V_{RHE}$), the phase angle in the low-frequency region remarkably increased, and the semicircle in the Nyquist plots appeared inside the

negative coordinate system. This is a typical catalyst poisoning phenomenon of Pt-based catalysts[21,29,30]. The $R_{ct}$ calculated from the Nyquist plots also increased sharply when the potential reached 0.8 $V_{RHE}$, while the solution resistance ($R_s$) showed no noticeable changes (Fig. 1c). This indicates that the main reason for the poisoning phenomenon was $R_{ct}$, which originated from the interface between the catalyst surface and the electrolyte.

Furthermore, the poisoning phenomenon was not solely due to the formation of poisonous species; changes in the catalyst can also restrict GLY adsorption, leading to a decrease in current during prolonged electrolysis. Thus, an open-circuit potential (OCP) test was conducted to evaluate GLY adsorption before and after various electrolysis durations at 0.7 $V_{RHE}$ (Fig. 1d). After the injection of 50 mM GLY, the OCP changes decreased from 0.33 to 0.26 $V_{RHE}$ as the electrolysis time increased, indicating weaker GLY adsorption, possibly caused by the oxidation of the catalyst during the electrolysis process. The OCP results confirmed that the poisoning phenomenon, which limited the adsorption of GLY on the catalyst surface, was not only due to the accumulation of glycerol oxidation intermediates[31], but also caused by changes in the catalyst during the electrolysis process. The slow desorption rate of intermediates led to further deep oxidation and C−C cleavage. Therefore, the poisoning phenomenon of Pt-based catalysts during GEOR can result in two significant undesirable consequences: reduced current density and poor selectivity for C3 products (Fig. 1e).

### Electrocatalytic performance of GEOR with PE

PE is considered a simple and efficient strategy for dynamically reconstructing the catalyst surface and redistributing surface-adsorbed species during the electrocatalysis process[7]. Therefore, we propose using PE with a square-wave potential to reduce the coverage of $OH_{ad}$ and the accumulation of intermediates, thereby releasing active sites, while also preventing the oxidation of platinum catalysts (Supplementary Fig. 4). As shown in Supplementary Fig. 5a, the current densities of CE (dashed line) and PE ($E_L = 0.3$ $V_{RHE}$, $t_{EL} = t_{EH} = 0.5$ s; solid line) at different potentials were recorded. The current densities of CE decreased dramatically with an increase in electrolysis time, retaining only ≤10% of the current after 4000 s of electrolysis. Meanwhile, the current density retention of PE remained at 30–60%, which is significantly higher than that of CE (Fig. 2a and Supplementary Fig. 5). Additionally, even though PE only takes half the actual electrolysis time, its overall current density is still higher than CE. This implies that PE can effectively mitigate the catalyst poisoning phenomenon and offers the possibility of stable electrolysis over an extended period.

The product distributions of GLY oxidation based on CE and PE were further detected and calculated through high-performance liquid chromatography. The selectivity of GLA decreased from 41.7% to 27.2% as the potential of CE increased from 0.6 to 1.1 $V_{RHE}$ (Fig. 2b). When PE ($E_L = 0.3$ $V_{RHE}$, $t_{EL} = t_{EH} = 0.5$ s) was applied for GLY oxidation, the selectivity of GLA significantly increased from 37.1% to 53.7% (Fig. 2c). Moreover, the C1 and C2 products (oxalic acid, GA, and formic acid) increased with increasing oxidation potential, regardless of whether the electrolytic mode was CE or PE. Since the Coulomb amount stays the same (~20 C) for different electrolytic conditions, the corresponding GLY conversion rate tends to decrease with the increase of C1&C2 products, which is due to the fact that more electricity is used for C-C bond breaking. GLAD, an important intermediate, was not observed in the glycerol oxidation products, because GLAD can undergo spontaneous base-catalyzed dehydration and Cannizzaro rearrangement in a strongly alkaline environment, ultimately transforming into LA[32]. This suggests that the main byproduct of GEOR was LA derived from GLAD. The LA content during CE was higher than that during PE, indicating that GLAD accumulated on the catalyst surface during prolonged electrocatalysis. As the consumption of surface

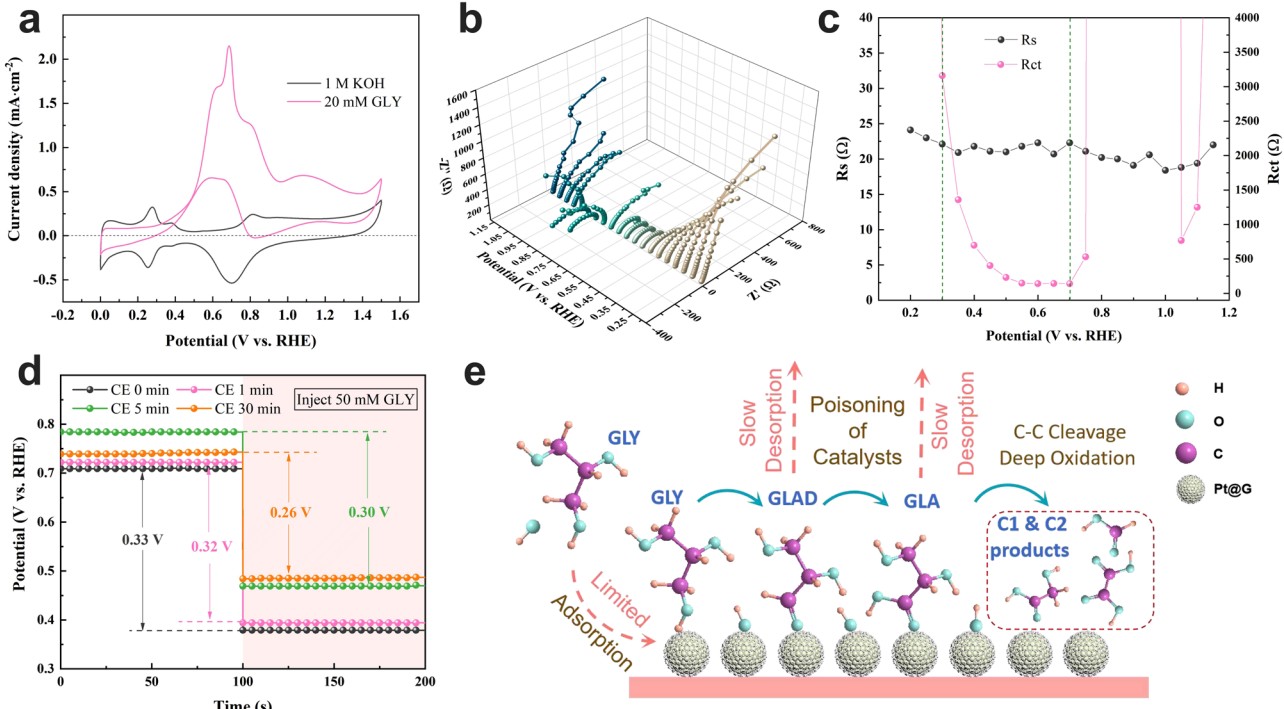

**Fig. 1 | Limitations of CE for the GEOR on Pt@G. a** CV curves of Pt@G in 1 M KOH with and without 20 mM GLY. **b** Nyquist plots and (**c**) corresponding charge-transfer resistance ($R_{ct}$) and solution resistance ($R_s$) at various potentials. **d** Open-circuit potential curves of Pt@G after CE with different times in KOH, with GLY being injected subsequently. The numbers in the figure indicate the value of the potential (vs. RHE) difference of the OCP. **e** Schematic of CE-based GEOR over Pt@G. GLY glycerol, GLAD glyceraldehyde, GLA glyceric acid.

$OH_{ad}$ and the occupation of active sites by poisonous species hindered the re-adsorption of $OH_{ad}$, GLAD could not be rapidly further oxidized to GLA. The results demonstrate that this situation can be efficiently alleviated by the pulse potential strategy.

To further enhance the selectivity of GLA, GEOR experiments were conducted in electrolytes with different pH values (Supplementary Fig. 6). The selectivity of GLA initially increased and then decreased as the pH increased, with the optimal electrolyte being 1 M KOH. The lower selectivity of GLA in 3 M KOH was due to the rapid Cannizzaro rearrangement on the catalyst surface. Considering both GLA selectivity and current density, the potentials for PE with $E_L = 0.3$ $V_{RHE}$ and $E_H = 0.7$ $V_{RHE}$ were chosen for further condition optimization.

The charge and conversion rate increased as the proportion of $t_{EH}$ increased (Supplementary Fig. 7a, b). Conversely, the selectivity for C3 and GLA decreased as the proportion of $t_{EH}$ increased, with the highest values of 100% and 86.6% achieved at $t_{EL} = 2$ s and $t_{EH} = 0.05$ s (Supplementary Fig. 7c and Fig. 2d). The lowest LA selectivity occurred when $t_{EH}$ was set to 0.05 s (Supplementary Fig. 7d). This suggests that the high selectivity for C3 and GLA likely originated from lower concentrations of GLAD and GLA on the catalyst surface, which can prevent LA formation through spontaneous reactions and inhibit the deep oxidation of GLA to produce low-carbon products. However, having a too-low $t_{EH}/t_{EL}$ ratio was also detrimental to the reaction rate, resulting in the lowest GLY oxidation current density (Supplementary Fig. 8–10). Therefore, the optimal PE condition in this study was $E_L = 0.3$ $V_{RHE}$, $E_H = 0.7$ $V_{RHE}$, $t_{EL} = 0.5$ s, and $t_{EH} = 0.05$ s (Supplementary Fig. 11, PE3). To further enhance the selectivity of GLA, reducing the local GLY concentration on the electrocatalyst surface is an effective method owing to the lower LA production (Supplementary Fig. 12). Under this condition (PE3) with 20 mM GLY, 87.3–84.6% C3 selectivity and 81.8–74.4% GLA selectivity were achieved, along with a 14.7–59.4% GLY conversion rate (Fig. 2e and Supplementary Fig. 13). This performance surpasses that of most reported results in the literature (Fig. 2f and Supplementary Table 3)[24,33–38].

## Effect of electrolytic mode on physical and chemical properties Pt@G during GEOR

To investigate the impact of CE and PE modes on the electrocatalyst, the physical and chemical properties of Pt@G before and after electrolysis were examined. First, X-ray diffraction (XRD) patterns and Raman analysis were conducted to gather structural information about Pt@G. Pt@G exhibited the typical polycrystalline Pt structure, with the main exposed crystal surface being Pt (111) (Fig. 3a). A peak corresponding to the graphite layer also occurred at $2\theta = 24°$. Raman spectra further confirmed the presence of the graphite layer (Fig. 3b). High-resolution transmission electron microscopy (TEM) was employed for a detailed analysis of morphology and structure (Fig. 3c, d). The Pt@G catalyst exhibited uniform nanoparticles with an average diameter of 2.4 nm. Additionally, graphite shells encapsulating Pt nanoparticles were observed (Fig. 3d, e), confirming the successful synthesis of Pt@G catalysts. The graphite shells encapsulating the platinum nanoparticles have defects and are not complete so that the platinum particles can act as catalysts. The selected-area electron diffraction (SAED) patterns (The inset figure in Fig. 3d) revealed that Pt@G possessed a polycrystalline structure with crystal surfaces of (111), (200), (220), and (311), which is consistent with the XRD analysis results.

Furthermore, the Pt@G catalysts subjected to PE3 and CE were further characterized via TEM and X-ray photoelectron spectroscopy (XPS). The TEM images and corresponding SAED patterns showed that the morphology and crystalline phase of Pt@G did not undergo significant changes after the PE3 and CE processes (Supplementary Fig. 14). However, according to the XPS results, the oxygen element content of PE3 applied for GLY oxidation (20.97%) was not significantly higher than that of the initial Pt@G (19.56%) (Supplementary Fig. 15 and Supplementary Table 4). XPS refined analysis revealed that the Pt (II) content increased from 16.2% to 28.6% after CE, while the Pt (II) content of the catalyst used for PE3 (17.3%) was not significantly higher than that of the initial catalyst (Fig. 4f–h and Supplementary Table 5).

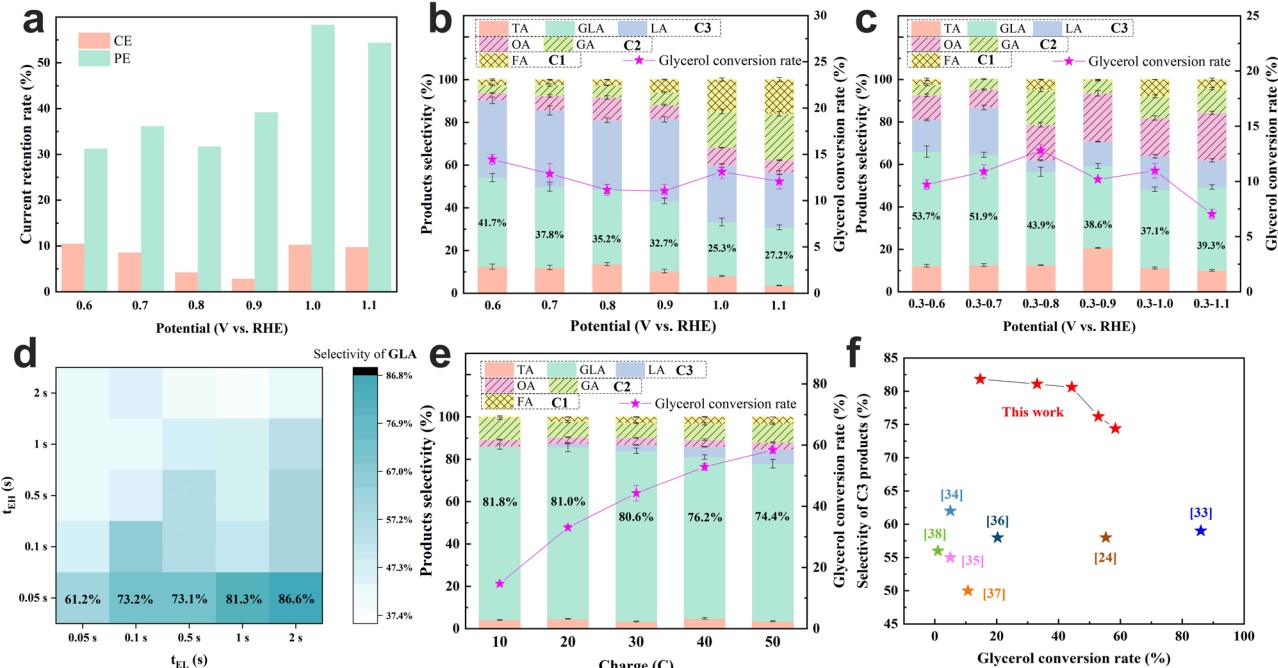

**Fig. 2 | Electrocatalytic performance of GEOR under the electrolytic mode of CE and PE. a** The current retention rate of CE and PE protocol with different potentials after 4000 s electrolysis. Product distribution and conversion rate of GEOR under the electrolytic modes of (**b**) CE and (**c**) PE ($E_L$ = 0.3 $V_{RHE}$ for 0.5 s and $E_H$ = 0.6–1.1 $V_{RHE}$ for 0.5 s). The electrolytic charge of different electrolytic modes is basically kept at about 20 C. **d** Condition optimization of $t_{EL}$ and $t_{EH}$ ($E_L$ = 0.3 $V_{RHE}$, $E_H$ = 0.7 $V_{RHE}$, 50 mM GLY in 1 M KOH) with the selectivity of GLA and the electrolysis time is 1 h. PE1 – PE5: $E_H$ = 0.7 $V_{RHE}$ for 0.05 s, $E_L$ = 0.3 $V_{RHE}$ for 2.0, 1.0, 0.5, 0.2 and 0.05 s, respectively. **e** Selectivity and GLY conversion rate with 20 mM GLY under the electrolytic mode of PE3. The percentages in black indicate the corresponding selectivity of GLA. **f** Comparison of the GLA selectivity and GLY conversion rate (Data derived from Fig. 2e) in this work with those of previous reports and the numbers in parentheses in the figure indicate the corresponding reference numbers. **b**, **c**, **d**, **e** TA tartronic acid, GLA glyceric acid, LA lactic acid, OA oxalic acid, GA glycolic acid, FA formic acid.

## Effect of electrolytic mode on surface micro-environment surrounding Pt@G during GEOR

Revealing the changes of surface species on catalysts during the electrocatalyst is essential to elucidate the reaction process. To obtain information about alterations in the surface micro-environment and adsorbed species during the GEOR process, in situ Fourier-transform infrared (FTIR) spectroscopy analysis was conducted. The FTIR spectra in the region of 1200–1000 cm$^{-1}$ corresponded to the adsorption sites on the catalyst for the $\nu$(C-O) bond of glycerol[25,39]. The 1114–1126 cm$^{-1}$ region corresponded to a bridging alkoxy bond formed from one primary alcohol group of glycerol and two metal surface atoms[39]. The band at 1053 cm$^{-1}$ corresponded to the $\nu$(C-O) bond formed between one metal surface atom and a primary alcohol group of glycerol[39]. As shown in Supplementary Fig. 17, these two regions exhibited negative peaks compared with the OCP spectrum, and the regions decreased as the potential increased from 0.3 to 1.1 $V_{RHE}$, indicating the dehydrogenation of glycerol's primary alcohol group adsorbed on the catalyst surface. The symmetric stretch $\nu$(OCO) of twofold carboxylate species appeared at 1400 cm$^{-1}$ (marked in magenta), consistent with previous reports[29,40,41]. The band at 1533 cm$^{-1}$ corresponded to $\nu$(OCO) in a bridging bidentate configuration (also marked in magenta) involving two Pt sites[40]. The band centered at 1590 cm$^{-1}$ corresponded to the (C=O)$_{acyl}$ stretch mode of adsorbed aldehyde species[29]. The intensity of these bonds related to glycerol oxidation intermediates increased with an increase in potential (Supplementary Fig. 17) and

electrolysis time (Fig. 4a and Supplementary Fig. 18b). The intensity of these regions under CE reached the maximum value faster at 1.1 $V_{RHE}$ (Supplementary Fig. 18b) than at 0.7 $V_{RHE}$ (Fig. 4a). These results demonstrate that higher potential led to the rapid formation and accumulation of glycerol oxidation intermediates. Furthermore, the increase in electrolysis time resulted in the continuous accumulation of intermediates, which was the origin of the toxicity phenomenon of the catalyst.

The surface hydrogen bonds associated with OH$^-$ and glycerol oxidation intermediates concentration were further investigated. The bands around 3236 and 3541 cm$^{-1}$ were identified as surface OH groups involved in hydrogen bonds on the metal catalyst surface and free OH groups, respectively[40]. The bands in the 3100–3700 cm$^{-1}$ region increased as the potential increased (Supplementary Fig. 17) and with an increase in electrolysis time during CE at 0.7 $V_{RHE}$ (Supplementary Fig. 19) and 1.1 $V_{RHE}$ (Supplementary Fig. 18a). Additionally, the band at 1644 cm$^{-1}$, assigned to intermolecular hydrogen bonds[25], also displayed the same trends in changes (Fig. 4a, Supplementary Figs. 17 and 18b). There were two possible reasons for the increase in hydrogen bonds on the catalyst surface. First, as the potential increased, the surface OH$^-$ concentration also increased because the negative charge favored alkoxide formation and adsorption on the catalyst surface[40]. Second, a higher pH environment promoted the formation of aldehyde intermediates into a geminal diol structure ($C_2H_5O_2CH(OH)_2$)[42]. The geminal diol structure on Pt (111) exhibited stronger adsorption energy than GLY and GLA, indicating that the desorption of geminal diol from the catalyst surface into the solution was more challenging (Fig. 4d). Furthermore, the formation of this structure can facilitate the rapid conversion of GLAD into GLA, thereby improving the selectivity of GLA. Therefore, the intensity enhancement observed in the 3100–3700 cm$^{-1}$ region and the 1644 cm$^{-1}$ band

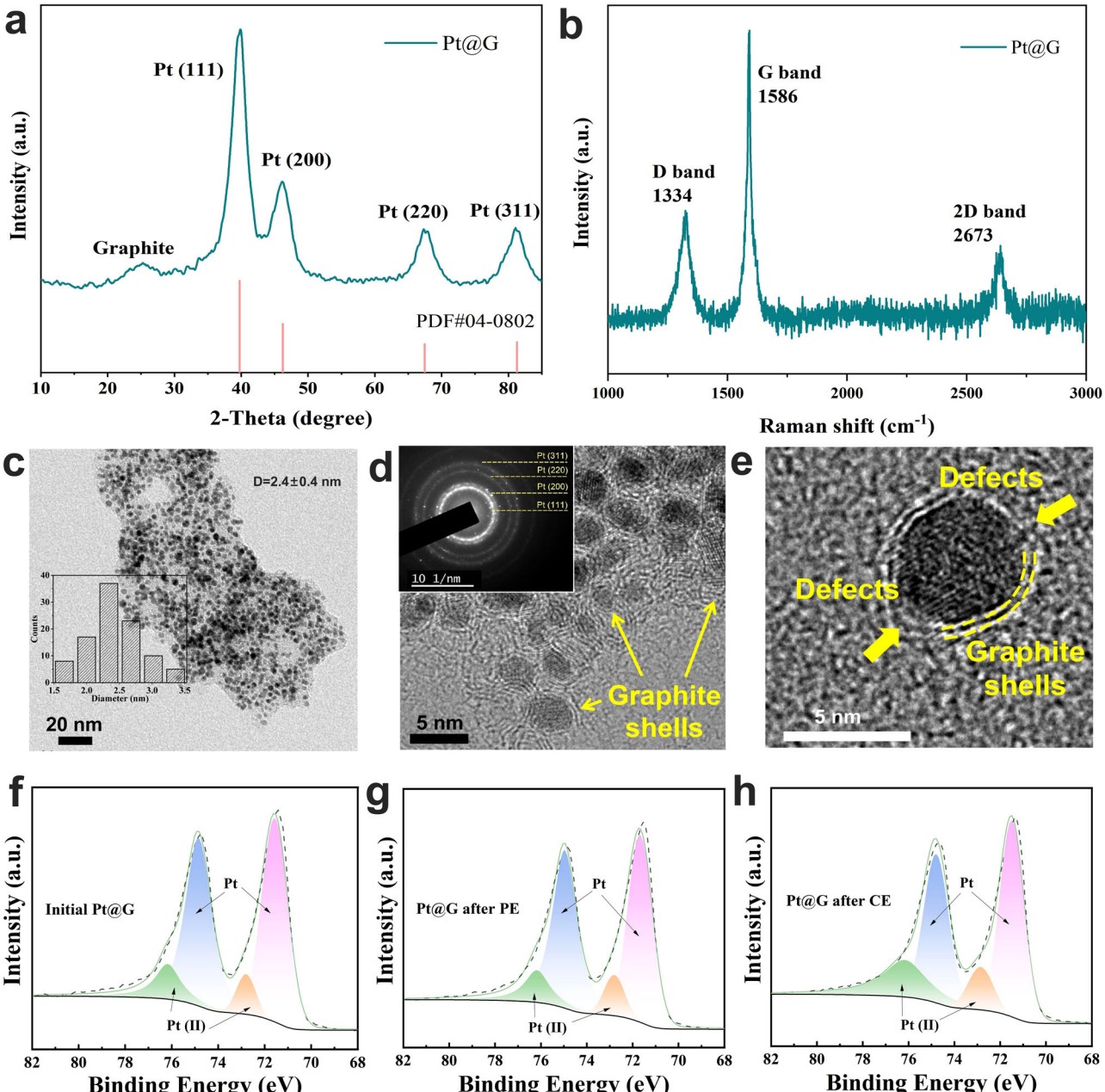

**Fig. 3 | Physical and chemical characterization of Pt@G. a** XRD pattern; (**b**) Raman spectrum; (**c–e**) TEM images of Pt@G. **c** Inset shows the particle size distribution of Pt@G nanoparticles. **d** Inset shows the SAED patterns of Pt@G. High-resolution X-ray photoelectron spectroscopy refined spectra of Pt 4$p$ for (**f**) initial Pt@G, (**g**) Pt@G used for PE3, and (**h**) Pt@G used for CE at 0.7 V$_{RHE}$.

with an increase in potential and electrolysis time provides evidence of the accumulation of OH$^-$ and glycerol oxidation intermediates during the CE process.

To further evaluate the effect of pulsed electrolysis on surface species during GEOR, in situ FTIR spectra of Pt@G electrodes with electrolytic modes of CE and PE were compared. The PE3 and CE at 0.7 V$_{RHE}$ with varying times were investigated (Fig. 4b and Supplementary Fig. 20). The intensity of bonds at 1400, 1588, and 1649 cm$^{-1}$ and the 3200–3600 cm$^{-1}$ region increased with time. However, when the PE3 protocol was applied, the intensity of these bands decreased, especially for the bonds at 1400 and 1588 cm$^{-1}$, which corresponded to carboxylate and aldehyde species, respectively. Furthermore, during the different PE protocols (Fig. 4c), these bonds decreased as t$_{EL}$ increased from 0.05 to 2 s (from PE5 to PE1). Moreover, the intensity of bonds at 1115 and 1058 cm$^{-1}$, associated with the primary alcohol

groups of glycerol, increased as t$_{EL}$ increased from 0.05 to 2 s. This indicates that the application of the PE method prevented the accumulation of toxic intermediate species on the electrode surface, promoted the rapid desorption and diffusion of carboxylate species, and freed up active sites for the re-adsorption of GLY. The combination of these effects was the key reason for the high GLA selectivity observed in GEOR based on the PE process.

**Finite element simulation and mechanistic understanding of PE for GEOR**

To more deeply elucidate the adsorption/desorption behaviors of surface species during the GEOR process, computer simulations were conducted using COMSOL Multiphysics software (detailed parameters can be found in the Methods section). During the electrocatalysis process, the rate of electron transfer primarily depends on the applied

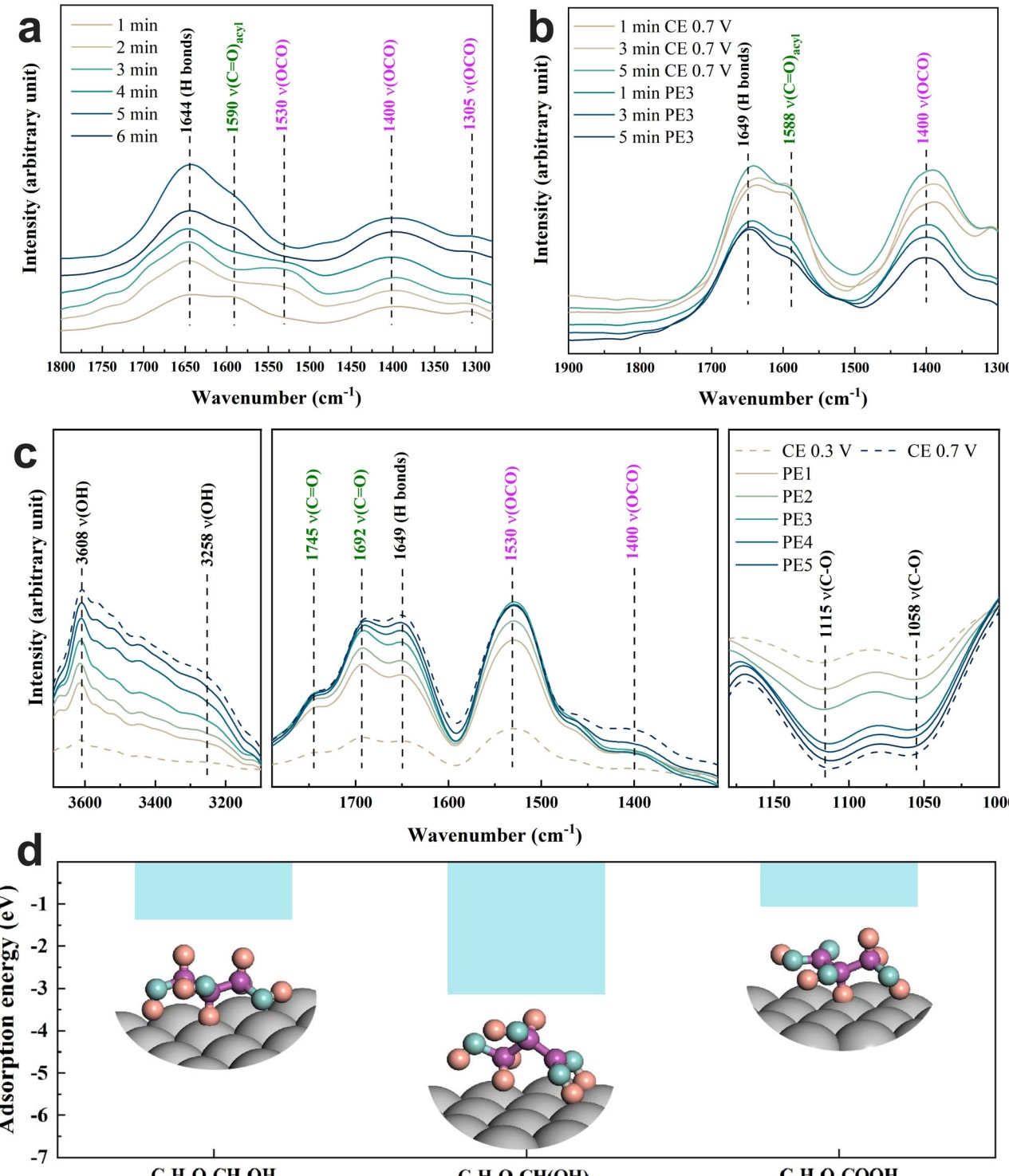

**Fig. 4 | In situ FTIR spectra of Pt@G catalysts in 1 M KOH with 50 mM GLY under different conditions. a** CE at 0.7 $V_{RHE}$ with different times, (**b**) PE3 after CE at 0.7 $V_{RHE}$ with different times and (**c**) different PE protocols (PE1 − PE5: $E_H = 0.7 \ V_{RHE}$ for 0.05 s, $E_L = 0.3 \ V_{RHE}$ for 2.0, 1.0, 0.5, 0.2 and 0.05 s, respectively). **d** Adsorption energy of GLY, $C_2H_5O_2CH(OH)_2$ (transformed from GLAD), and GLA on Pt (111) in an alkaline electrolyte. The three inset images present the optimized adsorption geometry of the corresponding three species on the Pt (111) surface. The color for each element is tangerine yellow for H, light blue for O, purple for C, and gray for Pt, respectively.

potential and the concentration of reactants and products within the electric double layer (EDL)[43]. Changes in potential also significantly influence the distribution of reactants and products owing to the effects of the electric fields within the EDL, particularly for electrically charged ions. Therefore, the effect on the distribution of $K^+$, $OH^-$, $H^+$, and $C_2H_5O_2COO^-$ within the EDL at different potentials was evaluated, without considering electrochemical/chemical reactions and adsorption. The concentration of $C_2H_5O_2COO^-$ on the catalyst surface (at a distance of 0 nm) was higher than that in the solution, and this difference became more pronounced as the potential increased from 0.1

to 0.7 $V_{RHE}$ (Fig. 5a). The distribution of $OH^-$ on the electrode surface exhibited the same trend, while the positive ions ($K^+$ and $H^+$) exhibited an opposite pattern (Supplementary Fig. 21). This suggests that negative ions ($OH^-$ and $C_2H_5O_2COO^-$) tended to accumulate on the electrode surface owing to the influence of electric fields. Conversely, when a relatively lower potential was applied, the desorption and diffusion of negative ions were promoted.

Finite element analysis was conducted to simulate the local concentration changes of different species with the PE and CE processes, and the adsorption/desorption of reactants and products and electrochemical and chemical reactions were considered. The time-dependent local concentration profile of GLAD is shown in Fig. 5b. At a constant potential of 0.7 $V_{RHE}$, GLAD accumulated on the catalytic surface owing to its strong adsorption onto the Pt site. The surface local GLA concentration exhibited the same trend (Supplementary Fig. 22). In contrast, the application of PE effectively mitigated the enrichment of GLAD and GLA, with local concentrations decreasing as $t_{EL}$ increased (from PE5 to PE1). These results align with the results of previous analyses (Fig. 4b, c). The visualized 2D concentration profiles of GLAD with PE3 are presented in Fig. 5c, offering a visual representation to further comprehend the effect of pulse potential on the accumulation and conversion of electrode species.

In addition to the local concentration, the surface coverage of different species directly affected electron transfer in the electrocatalyst reaction. At a constant potential of 0.7 $V_{RHE}$, the coverage of $OH_{ad}$ (Supplementary Fig. 23a) and GLY (Supplementary Fig. 23b) decreased significantly owing to their consumption and the accumulation of poisoning species that occupy a large number of active sites (Supplementary Fig. 23c, d). $OH_{ad}$ played an essential role in GEOR. Decreased $OH_{ad}$ coverage resulted in a large number of intermediate species being unable to transform rapidly, leading to rapid catalyst deactivation and a low current density. In contrast, the application of PE resulted in lower coverage of GLAD and GLA and higher coverage of $OH_{ad}$ and GLY. This demonstrates that the rapid oxidation of GLAD, due to higher OH coverage and fast diffusion of GLA with lower potential ($E_L$), promoted the refreshing of active sites for the re-adsorption of $OH^-$ and GLY. Moreover, the short pulse time of high potential effectively prevented further GLA oxidation (Fig. 5d). These strategies provide an effective means to prevent catalyst deactivation and achieve high C3 selectivity during GEOR.

## Discussion

We developed a PE strategy to address the deactivation issue of Pt-based electrocatalysts by altering the catalyst surface microenvironment. In situ characterization and computer simulations revealed that the PE electrolytic mode prevented the over-accumulation of poisoning intermediate species, promoted the desorption and diffusion of the target product (GLA) into the solution, and accelerated the re-adsorption of $OH_{ad}$ and GLY. Additionally, the application of the PE protocol alleviated Pt oxidation during the CE process, further enhancing GLY adsorption. Consequently, remarkable C3 selectivity (87.3–84.6%) and GLA selectivity (81.8–74.4%) were achieved. This work provides a simple and efficient strategy to prevent the deactivation of noble-based electrocatalysts and obtain high-value C3 products from glycerol with high selectivity.

## Methods
### Preparation of Pt@G
The graphitic Pt nanocrystals (Pt@G) were prepared in a chemical vapor deposition (CVD) system[44,45]. In a typical procedure, the fumed silica (1.0 g) was impregnated with $H_2PtCl_6 \cdot 6H_2O$ (50 mg) in 100 mL methanol and sonicated for 2 h. The mixture was then dried at 60 °C, and the powder was placed into a methane CVD chamber for graphitic growth with a methane flow of 150 $cm^3 \cdot min^{-1}$ and hydrogen flow of 20 $cm^3 \cdot min^{-1}$ for 5 min at 1000 °C. After that, the sample was etched

with HF and $HNO_3$ to dissolve the silica, then washed and centrifuged to obtain the Pt@G catalyst.

### Characterizations
The X-ray diffraction (XRD) analysis was conducted on a Siemens D500 diffractometer with a Cu Ka source (1.54056 Å). The Raman spectra were obtained with Renishaw's InVia Raman system with 633 nm laser excitation (Renishaw, UK). TEM images were taken with a JEM-2010 (JEOL, Japan). X-ray photoelectron spectroscopy (XPS) spectra were acquired with an ESCALAB 250Xi X-ray photoelectron spectrometer.

### Electrochemical measurements
The electrochemical test was carried out in a three-electrode system with the prepared titanium mesh loaded with catalysts (0.5 $mg \cdot cm^{-2}$) as the working electrode, Hg/HgO as the reference electrode and graphite rod as the counter electrode. The CHI 660E was used as the electrochemical workstation. LSV was performed with a scan rate of 5 $mV \cdot s^{-1}$ in the single cell. The electrolysis test was performed in the divided cell, separated by the Nafion 117 membrane. The pulsed potential electrolysis (PE) was conducted by using the muti-potential steps technique. The relative lower and higher potentials are named $E_L$ and $E_H$ respectively. The responding time is named $t_{EL}$ and $t_{EH}$. In situ electrochemical impedance spectroscopy (EIS) measurements were carried out by Autolab PGSTAT302N (Eco Chemie, Utrecht, the Netherlands) over the frequency range from $10^5$ to $10^{-2}$ Hz with an amplitude of 10 mV.

### Product analysis
The determination of electrolysis products was achieved using high-performance liquid chromatography (HPLC, LC 2030 C, Shimadzu, Japan) equipped with both a UV detector and a RID detector. The mobile phase was 5 mM aqueous $H_2SO_4$ with a flow rate of 0.5 $mL \cdot min^{-1}$ and the column temperature was 65 °C. The UV detector with 210 nm was used for the quantification of oxalic acid (OA), tartronic acid (TA), glyceraldehyde (GLAD), dihydroxyacetone (DHA), glyceric acid (GLA), lactic acid (LA) and glycolic acid (GA), formic acid (FA). The RID detector was used to determine the GLY concentration. The conversion rate of GLY and the selectivity of products were calculated as follows:

$$Conversion\ rate\ (\%) = \frac{n_{GLY,0} - n_{GLY,t}}{n_{GLY,0}} \times 100\% \tag{1}$$

$$Selectivety\ (\%) = \frac{n_{product,t} \times N}{3 \times (n_{GLY,0} - n_{GLY,t})} \times 100\% \tag{2}$$

where $N$ is the carbon atom number of the product.

### In situ FTIR study
The in situ FTIR was performed under different potentials by an infrared spectrometer (BRUKER, TENSOR II, USA) which was equipped with a mercury cadmium telluride (MCT) detector cooled by liquid nitrogen. The spectra were collected at a resolution of 4 $cm^{-1}$ in the 4000–750 $cm^{-1}$ range. The Au film was chemically deposited on the single-crystal silicon. First, the Si crystal was immersed in the mixture solution (HCl/$HNO_3$ = 3: 1, v/v) for 2 h to wash the surface contaminants. Then, the Si crystal surface was polished with 0.05 mm $Al_2O_3$ powder for 30 min and then washed with ethanol and deionized water. The clean Si crystal was further immersed in the mixture solution ($H_2SO_4$ (98%)/$H_2O_2$(30%) = 7/3, v/v) for 15 min at 55 °C and then washed with deionized water. Next, the crystal face of Si was immersed in a gold plating solution (6 $mmol \cdot L^{-1}$ $NaAuCl_4$) for 5 min at 55 °C followed by washing with deionized water and air dried. The Pt@G catalyst was dropped on Au film for the in situ FTIR test. The tests were performed in a two-compartment electrochemical cell. The Au film

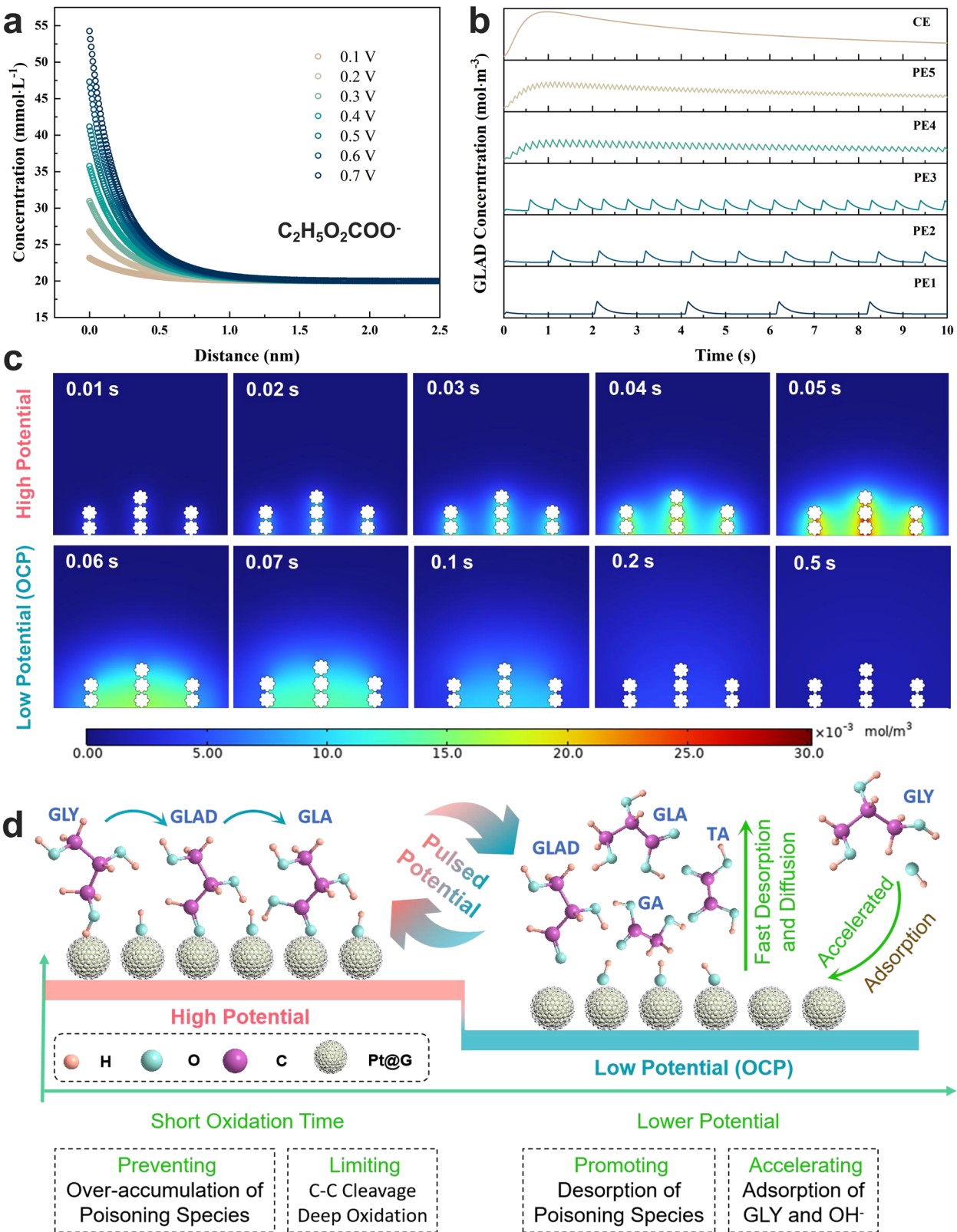

**Fig. 5 | Simulations of the GEOR process with the PE protocol. a** Effect of applied constant potential on the diffusion of $C_2H_5O_2COO^-$ on the electrode surface. **b** Surface concentration of GLAD at different electrolysis times for CE and PE protocol (PE1 − PE5: $E_H$ = 0.7 $V_{RHE}$ for 0.05 s, $E_L$ = 0.3 $V_{RHE}$ for 2.0, 1.0, 0.5, 0.2 and 0.05 s, respectively). **c** Snapshots of GLA at different electrolysis times for the PE3 protocol. **a**, **b**, **c** Data obtained through COMSOL multiple physical quantities. The catalyst of Pt@G is represented by seven round white dots. **d** Schematic of the pathways of PE-based GEOR. GLY glycerol, GLAD glyceraldehyde, GLA glyceric acid, TA tartronic acid, GA glycolic acid.

loaded with catalyst, graphite rod and Hg/HgO were used as working electrodes, counter electrode and reference electrode respectively.

## Adsorption energy calculation

Adsorption energy calculations were performed using the Vienna ab initio simulation package (VASP 5.4)[46]. The projector-augmented-wave (PAW) pseudopotentials and Perdew-Burke-Ernzerhof (PBE) functional were used for electron-ion interactions and exchange-correlation energy functional, respectively[47]. The kinetic energy cutoff for plane-wave basis was set to 500 eV cutoff was used for describing valence electrons. The van der Waals (vdW) interactions among organic molecules and surfaces of Pt were calculated by the DFT-D3 method[48]. Pt is a $(4 \times 4)$ supercell of the (111) surface. The force convergence criterion was set to −0.05 eV/Å and the energy convergence criterion was $10^{-5}$ eV. The Brillouin zone was sampled with a Gamma-centered k-point grid of $3 \times 3 \times 1$ for Pt supercells. A vacuum layer of 15 Å was adopted in the three models.

## Finite element simulations of the diffuse double layer

To evaluate the effect of potential changes on the distribution of the electrically charged ions during the electrocatalysis process. The model with a double-layer model combines the transport of diluted species and electrostatics physics interfaces to account for mass transfer was established by COMSOL Multiphysics 6.1 software. The model contains one charged electrode adjacent to the bulk solution.

The geometry of the model is defined by a single interval between 0 and L, indicating a one-dimensional structure. This interval encompasses the electrolyte phase spanning from the electrode through the diffuse double layer and extending to the electroneutral bulk solution. The boundary condition set at x = 0 is utilized to regulate the compact component of the double layer. The main parameters for finite element simulations of the diffuse double layer are shown in Supplementary Table 1.

## Finite element simulations of the GEOR

The modelling of the GEOR with pulsed potential electrolysis process was performed in COMSOL Multiphysics 6.1 software with a diluted species transport module combined with secondary current distribution. Models of electrode geometries including planar and electrode with nanoparticles were built. The domain equation is the diffusion equation, known as Fick's 2nd law, which describes the chemical transport of the electroactive species. known as Fick's 2nd law, which describes the chemical transport of the electroactive species.

$$\nabla \times (D_i \nabla c_i) = 0 \tag{3}$$

At the bulk boundary, a uniform concentration was assumed to be equal to the bulk concentration for the reactants, where the products have zero concentration. At the electrode boundary, the reactant species are reduced to form the products. In this study, the GEOR process on the electrode surface was divided into two processes including the adsorption/desorption reactions (without the charge transfer) and the electrochemical redox reactions (with the charge transfer). Meanwhile, there are two equilibrium reactions in the solution.

Adsorption/desorption reactions (without the charge transfer):

$$OH^- + ^* \leftrightarrow OH_{ads}^- \tag{4}$$

$$C_2H_5O_2CH_2OH + ^* \leftrightarrow C_2H_5O_2CH_2OH_{ads} \tag{5}$$

$$C_2H_5O_2CHO + ^* \leftrightarrow C_2H_5O_2CHO_{ads} \tag{6}$$

$$C_2H_5O_2COOH + ^* \leftrightarrow C_2H_5O_2COOH_{ads} \tag{7}$$

Electrochemical redox reactions (with the charge transfer):

$$C_2H_5O_2CH_2OH_{ads} + 2OH_{ads}^- \rightarrow C_2H_5O_2CHO_{ads} + 2H_2O + 2^* + 2e^- \tag{8}$$

$$C_2H_5O_2CHO_{ads} + 2OH_{ads}^- \rightarrow C_2H_5O_2COOH_{ads} + H_2O + 2^* + 2e^- \tag{9}$$

Equilibrium reactions:

$$C_2H_5O_2COOH \leftrightarrow C_2H_5O_2COO^- + H^+ \tag{10}$$

$$H_2O \leftrightarrow OH^- + H^+ \tag{11}$$

where the * stands for the free active sites. For the adsorption of different spices expressed as $A_{ads}$.

The local current of the electrochemical reaction is described in the following expressions:

$$i = i_0 \left( C_R \times \exp\left(\frac{n\alpha_a F\eta}{RT}\right) - C_O \times \exp\left(\frac{-n\alpha_c F\eta}{RT}\right) \right) \tag{12}$$

$$i_0 = k_0 \times F \times \gamma \tag{13}$$

where $C_R$ and $C_O$ are the expressions for the concentration of reductant and oxides; $\alpha_a$ and $\alpha_c$ are the electron exchange coefficient, $F$ is the Faraday constant, and $R$ is the ideal gas constant. $T$ is temperature. $n$ is the electron transfer number. $\eta$ is the overpotential of the reaction. $k_O$ is the kinetic rate constant. $\gamma$ is the in-situ density of the absorption spices. The main parameters for finite element simulations of the GEOR are shown in Supplementary Table 2.

## Data availability

All data are available in the manuscript, the supplementary materials, and from the authors on request. Source data are provided as a Source Data file. Source data are provided with this paper.

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

## Acknowledgements

Y.Z. acknowledges the support from the National Key R&D Program of China (2022YFA1504200), the National Natural Science Foundation of China (22122901), and the Provincial Natural Science Foundation of Hunan (2021JC0008, 2021JJ20024, 2021RC3054). H.P. acknowledges the support from the National Natural Science Foundation of China (52371240).

## Author contributions

Y.Z., H.P., and S.W. conceived and supervised the project. W.C. conducted experiments and analyzed data. L.Z. performed STEM measurement and analysis. L.X. performed the DFT calculations and prepared the DFT section for the manuscript. Y.H. conducted some experiments. W.C. performed COMSOL finite element simulations and wrote the paper.

## Competing interests

The authors declare no competing interests.
