## [Peer Review File · Nature Communications]

REVIEWER COMMENTS

Reviewer #1 (Remarks to the Author):

The manuscript is well written and the objectives are clear. Interesting and convincing interpretation of the FTIR-data.

Below are some comments that need to be clarified:

1. Unclear why the Pt particles shall be covered by graphite. I interpret the writing Pt@C so that the authors have a core of Pt and a shell of graphite after synthesis. I thought Pt particles should just be in good contact with graphite and that Pt is the active catalyst and not graphite. I mean the active catalyst shall face the electrolyte. Do the authors mean that graphite is the active catalyst? This must be clarified otherwise it is very difficult to understand the materials.
2. Figure 3d: It is not clear to me from the figure how the Pt particles are encapsulated by graphite shells as the term Pt@C means. Replace this figure with one obtained at a higher magnification so the relation in between Pt and graphite becomes clear to the reader.
3. Figure 2a contains too much information. It is difficult to relate the curves to the correct label.
4. Page 11, line 208: The average nanoparticle diameter is given with too high accuracy.
5. The selectivity of different oxidation products is described, but the conversion rate of glycerol is less discussed in the text. It is said in the caption to Figure S6 that it contains GLY conversion rate but it shows selectivity. However, Figure 2f and Figures S11 and S12 contain conversion rate.
6. Make it clearer to the reader how long current pulses are used.

Reviewer #2 (Remarks to the Author):

In the submitted manuscript, the authors report a pulsed electrocatalysis method to selectively oxidize glycerol (GLY) to glyceric acid (GLA) with a Pt-based catalyst. Under pulsed reaction conditions, the authors applied two different potentials (0.3 and 0.7 VRHE) for the glycerol electro-oxidation reaction. When the potential of 0.7 VRHE is applied, the glycerol can rapidly be oxidized to glyceraldehyde, glyceric acid, and other products. However, keeping the potential in 0.7 VRHE for a long period of time results in the accumulation of poisoning intermediate species and the over-oxidation of catalysts. Thus, the relatively low potential (0.3 VRHE) was applied, which can facilitate the rapid desorption of GLA and release active sites on the catalyst surface enabling the re-adsorption of OH_{ad} and GLY. Furthermore, this approach effectively prevented the catalyst from reaching an over-oxidized state.

The authors present a facile method to regulate the adsorption/desorption species on the catalyst surface, simultaneously mitigating the deactivation of noble metal catalysts, and achieving high C₃ and glyceric acid. Overall, this work is well-structured, and of high significance to the field. So the manuscript merits publication in Nature Communications. However, there are some remaining remarks that require attention and addressing.

1. As the authors said, lactic acid is a major side product of glycerol electro-oxidation reaction which is derived from glyceraldehyde by Cannizzaro rearrangement. I was wondering how the authors to reduce lactic acid production, especially in an alkaline environment.
2. It was noted that the catalysts were loaded on the titanium mesh with relatively low current density. Why don't use the traditional carbon cloth or carbon paper as the collector of fluids for the larger current density? What is the difference between them? And the mass loadings of catalysts should be clarified.
3. Page 4, the expression of "low oxidation state" in line 66 is confusing.
4. Page 18, all the C₂H₅O₂COO⁻ should be C₂H₅O₂COO⁻.
5. The authors should mention the time or charge with pulsed or constant electrolysis in Fig.2 (b-

d).

6. Page 10, Line 180, "Supplementary Fig. 7 and 7b" should be "Supplementary Fig. 7a and 7b".

7. As the author claimed that "the Pt-based catalysts will be deactivated when a high potential (0.7 VRHE) is applied", why would the further deep oxidation and C-C cleavage still occur? Page 7, Line 122-129.

8. Why did the authors choose graphitic carbon encapsulated Pt nanoparticles as catalysts instead of regular Pt nanoparticles or commercial Pt/C?

9. In the article, the authors explored the effect of different oxidation potentials (high potential) on glycerol oxidation, but the reduction potentials (low potential) have not been mentioned. In other words, what is the reason the authors would choose 0.3 VRHE as the reduction potential?

10. As shown in Fig. 2b, at 0.6 VRHE of CE, the selective performance of GLA also can reach 41.7%. What are the main advantages of pulsed potentials?

Reviewer #3 (Remarks to the Author):

The authors present a strategy using pulsed potential for the electrocatalytic selective oxidation of glycerol to glyceric acid. In comparison to a constant potential, this approach facilitates the desorption of poisoning intermediates, glyceraldehyde (GLAD), and OH⁻, resulting in higher glyceric acid selectivity. Overall, the chosen research topic is interesting, and the manuscript is logically structured, making it suitable for publication. However, several issues need to be addressed:

i. While both FTIR spectra and COMSOL indicate consistent results, demonstrating a monotonic decrease in the adsorption of aldehyde species on the catalyst surface across experimental conditions PE1 to PE5. However, in experiments, PE3 has the best catalytic performance, not PE5. How to understand this trend and its relationship with electrocatalytic performance?

ii. According to the description in the Methods section, the reaction energy barriers in Fig. 4d are all simulated using the gas phase molecular model, although the solvation model is added as a correction. However, in the experiment, it is regarded as an electrocatalytic process that occurs on the surface of the catalyst, and the focus is also on the impact of poisoning intermediates on the catalyst surface. Therefore, there is an obvious gap between this simulation and the actual conditions.

iii. The atomic structure diagram of "C₂H₅O₂CHO" in Fig. 4e portrays two hydroxyl groups rather than aldehyde groups, contradicting the description as "hexylene glycol" in the text (lines 281-284, page 16). Furthermore, how to understand that the adsorption energy of the dihydroxyl structure is greatly enhanced compared to the single hydroxyl structure (nearly 2 eV stronger as shown in the figure), just because of the addition of one more hydroxyl group?

iv. The description of the catalysts, "graphitic Pt nanocrystals", lacks clarity (line 61, page 4). It is recommended to specify whether Pt particles are supported on graphitic carbon, capsuled in graphitic carbon, or embedded within the graphitic carbon layer.

v. Schematics (Fig. 1e and Fig. 5d) looks a bit confusing. The H atoms are sometimes large and sometimes small; the H of the OH group points the surface, but actually most of the time O points the surface; the functions of high potential and low potential are to promote the oxidation reactions and promote the desorption of poisoning species, respectively, which was not reflected clearly.

Minor issues:

i. In Fig. 2a, several lines are nearly invisible.

ii. The seven round white dots in each sub-plot in Fig. 5c lacks an explanation.

iii. In Fig. 2c, the C1 and C2 products should be labeled separately for improved clarity.

iv. In Fig. S23, does "DC" represent "constant potential"?

v. Formulas on Pages 23, 26, and 27 suffer from blurry fonts, making it challenging to assess their correctness.

RESPONSE TO REVIEWERS' COMMENTS

Reviewer #1:

The manuscript is well written and the objectives are clear. Interesting and convincing interpretation of the FTIR-data.

Below are some comments that need to be clarified:

1. Unclear why the Pt particles shall be covered by graphite. I interpret the writing Pt@C so that the authors have a core of Pt and a shell of graphite after synthesis. I thought Pt particles should just be in good contact with graphite and that Pt is the active catalyst and not graphite. I mean the active catalyst shall face the electrolyte. Do the authors mean that graphite is the active catalyst? This must be clarified otherwise it is very difficult to understand the materials.

Response: Thanks for the reviewer's comments. As described in the Methods section, the Pt particles with the graphite shell catalysts (Pt@G) were prepared in a chemical vapor deposition (CVD) system. In this study, the main role of the graphite shell is to maintain the dispersion of the Pt nanoparticles (preventing nanoparticle aggregation) and the stability of the catalyst (*CCS Chem.* 2022, 4, 2382–2395; *Nat. Commun.* 2021, 12, 2002). Meanwhile, the well-contacted graphene layer also facilitates electron transfer during catalysis.

As for determine the active site of the catalyst, we collected the CV curves of Pt@G and graphene in the region of 0 to 1.1 V_{RHE} in 1.0 M KOH after adding 50 mM GLY. As shown in Figure R1, the graphene catalyst had no observable oxidation current peaks, while the Pt@G showed significant oxidation current peaks around 0.7 V_{RHE}. It means that the Pt is the active source of the catalyst rather than graphitic carbon. In the meantime, the TEM analysis (Figure 3e) showed that the encapsulated graphite shell of the platinum nanoparticles is incomplete; many defects allow the platinum nanoparticles to bind to the substrate and undergo a catalytic reaction. We have added relevant discussion in the revised manuscript.

Revision:

Line 215-217, page 11: As shown in **Fig. 3e**, the graphite shells with two layers encapsulated the platinum nanoparticles. Meantime, the graphite shells have defects that allow the platinum nanoparticle core to bind to the substrate and undergo a catalytic reaction.

Figure R1. a) CV curves of graphene in 1 M KOH with and without 20 mM GLY. **b)** CV curves of graphene and Pt@G in 1 M KOH with 20 mM GLY.

2. Figure 3d: It is not clear to me from the figure how the Pt particles are encapsulated by graphite shells as the term Pt@C means. Replace this figure with one obtained at a higher magnification so the relation in between Pt and graphite becomes clear to the reader.

Response: Thank you for pointing out this issue. As shown in Fig. 3e, the higher magnification TEM image was added to the revised manuscript. It was shown that two layers of graphene wrapped around the outside of the platinum nanoparticle. Meanwhile, the defects were observed on the graphite shell, which provides a channel for the platinum catalyst to catalyze a reaction with the substrate. We have added relevant descriptions in the revised manuscript.

Revision:

Line 215-217, page 11: As shown in Fig. 3e, the graphite shells with two layers encapsulated the platinum nanoparticles. Meantime, the graphite shells have defects that allow the platinum nanoparticle core to bind to the substrate and undergo a catalytic reaction.

Fig. 3. Physical and chemical characterization of Pt@G. **a** XRD pattern; **b**) Raman spectrum; **c–e** TEM images of Pt@G, and insert of Fig. 3d is SAED patterns of Pt@G. High-resolution X-ray photoelectron spectroscopy refined spectra of Pt 4p for **f** initial Pt@G, **g** Pt@G used for PE3, and **h** Pt@G used for CE at 0.7 V_{RHE}.

3. Figure 2a contains too much information. It is difficult to relate the curves to the correct label.

Response: Thank you for the kind suggestion. In **Figure 2a**, we mainly compare the stability of the current of CE and PE with different potentials, which is an essential reflection of the stability of the catalyst. In order to show the change in the current retention rate more intuitively and clearly, we directly extracted the data of the current retention rate after 4000 s of electrolysis for comparison. We made a bar chart in **Figure 2a** in the revised manuscript (as shown in **Figure R2**). The corresponding I-t diagram is placed inside the Supporting information (**Figure S5**).

Figure R2. The current retention rate of CE and PE protocol with different potentials after 4000 s electrolysis.

4. Page 11, line 208: The average nanoparticle diameter is given with too high accuracy.

Response: Thanks for the reviewer's comments. The average nanoparticle diameter was recorded and calculated in this study with the software of ImageJ. We have reduced the adequate number of nanoparticle diameters to two digits of 2.4 nm, and the corresponding changes were highlighted in red in the revised manuscript (Line 213, page 11).

Revision:

Line 213, page 11: The Pt@G catalyst exhibited uniform nanoparticles with an average diameter of 2.4 nm.

5. The selectivity of different oxidation products is described, but the conversion rate of glycerol is less discussed in the text. It is said in the caption to Figure S6 that it contains GLY conversion rate but it shows selectivity. However, Figure 2f and Figures S11 and S12 contain conversion rate.

Response: Thank you for pointing out this issue. In order to compare the effect of different electrolysis conditions on the oxidative selectivity of glycerol, we kept the Coulomb amount essentially the same (~20 C) for Figure S6, Figure 2b and 2c, thus resulting in a slight change in glycerol conversion under different electrolysis conditions. In general, there is a tendency for it to decrease with an increase in the C1&C2 products, and the corresponding discussion has been added and highlighted in the revised manuscript (Line 163-166, page 9). The GLY conversion rate data were added to Figure S6 in the revised supporting information (as shown in Figure R3).

Figure R3. The selectivity and GLY conversion rate with 50 mM GLY of PE ($E_L = 0.3 V_{RHE}$, $E_H = 0.7 V_{RHE}$, $t_{EL} = 0.5$ s, $t_{EH} = 0.05$ s) in different electrolytes with a same charge of 20 C.

Revision:

Line 163-166, page 9: Since the Coulomb amount stays the same (~20 C) for different electrolytic conditions, the corresponding GLY conversion rate tends to decrease with the increase of C1&C2 products, which is due to the fact that more electricity is used for C-C bond breaking.

6. Make it clearer to the reader how long current pulses are used.

Response: Thanks for the reviewer's comments. As stated in the previous question, to compare the effect of different electrolysis conditions on the oxidative selectivity of glycerol, we kept the charge essentially the same (~20 C) for Figure S6, Figure 2b, and 2c. Therefore, the electrolysis time is not the same, which is related to the amount of current density. As for the condition optimization of t_{EL} and t_{EH} (Figure 2d), we conducted the electrolysis for 1 hour. Corresponding descriptions are given in the caption of the figure as well as in the corresponding place in the revised manuscript and highlighted with red.

Revision:

Line 149-150, page 8: The electrolytic charge of different electrolytic modes is basically kept at about 20 C.

Line 151-152, page 8: Condition optimization of t_{EL} and t_{EH} ($E_L = 0.3 V_{RHE}$, $E_H = 0.7 V_{RHE}$, 50 mM GLY in 1 M KOH) with the selectivity of GLA and the electrolysis time is 1 hour.

Reviewer #2:

In the submitted manuscript, the authors report a pulsed electrocatalysis method to selectively oxidize glycerol (GLY) to glyceric acid (GLA) with a Pt-based catalyst. Under pulsed reaction conditions, the authors applied two different potentials (0.3 and 0.7 VRHE) for the glycerol electro-oxidation reaction. When the potential of 0.7 VRHE is applied, the glycerol can rapidly be oxidized to glyceraldehyde, glyceric acid, and other products. However, keeping the potential in 0.7 VRHE for a long period of time results in the accumulation of poisoning intermediate species and the over-oxidation of catalysts. Thus, the relatively low potential (0.3 VRHE) was applied, which can facilitate the rapid desorption of GLA and release active sites on the catalyst surface enabling the re-adsorption of OH_{ad} and GLY. Furthermore, this approach effectively prevented the catalyst from reaching an over-oxidized state.

The authors present a facile method to regulate the adsorption/desorption species on the catalyst surface, simultaneously mitigating the deactivation of noble metal catalysts, and achieving high C₃ and glyceric acid. Overall, this work is well-structured, and of high significance to the field. So the manuscript merits publication in Nature Communications. However, there are some remaining remarks that require attention and addressing.

1. As the authors said, lactic acid is a major side product of glycerol electro-oxidation reaction which is derived from glyceraldehyde by Cannizzaro rearrangement. I was wondering how the authors to reduce lactic acid production, especially in an alkaline environment.

Response: Thanks for the reviewer's comments. As we said, the glyceraldehyde can be transformed to lactic acid in a strong alkaline environment by Cannizzaro rearrangement. **Figure S6** shows that higher OH concentrations (3 M KOH) accelerate this reaction, leading to an increase in lactic acid. Meanwhile, the high concentration of glycerol (**Figure S12**) and the long pulse oxidation time (**Figure 2d** and **Figure S7**) means that there will be a high concentration of glyceraldehyde on the surface of the catalyst, which favors the production of lactic acid over glyceric acid. Therefore, in this work, we optimized the pH value of the electrolyte, the concentration of the substrate, and the duration of the oxidation potential in pulsed electrolysis with a view to improving the selectivity of glyceric acid rather than lactic acid.

2. It was noted that the catalysts were loaded on the titanium mesh with relatively low current density. Why don't use the traditional carbon cloth or carbon paper as the collector of fluids for the larger current density? What is the difference between them? And the mass loadings of catalysts should be clarified.

Response: For pulse catalysis, the repeated change in potential also causes a charging and discharging process of the double electronic layer, and this change does not affect the catalytic reaction. Therefore, to minimize the effect of the charging and discharging of the double electronic layer on the current of the catalytic reaction, we

have used a titanium mesh as a collector instead of carbon cloth or carbon paper. The mass loading of catalysts was 0.5 mg/cm^2 , added in the method section in the revised manuscript and highlighted in red.

Revision:

Line 397, page 22: The electrochemical test was carried out in a three-electrode system with the prepared titanium mesh loaded with catalysts ($0.5 \text{ mg}\cdot\text{cm}^{-2}$) as the working electrode.

3. Page 4, the expression of “low oxidation state” in line 66 is confusing.

Response: Thanks for the reviewer’s comments. In this sentence we mainly want to express that pulsed electrocatalysis prevents the over-oxidation of Pt catalysts. As shown in Figure 4(f-h) and Table 5, the XPS refined analysis reveals that the Pt(II) content increased from 16.2% to 28.6% after the CE, while the Pt(II) content of the catalyst after PE3 (17.3%) has no noticeable increase. We have revised the sentence in the revised manuscript and highlighted it in red.

Revision:

Line 64-66, page 4: Preventing the catalyst from reaching an over-oxidized state and releasing active sites on the catalyst surface enables the re-adsorption of OH_{ad} and GLY.

4. Page 18, all the $\text{C}_2\text{H}_5\text{O}_2\text{COO}^-$ should be $\text{C}_2\text{H}_5\text{O}_2\text{COO}^-$.

Response: Thanks for the reviewer’s comments. The corresponding revisions have been added to the revised manuscript and were highlighted in red.

Revision:

Line 322, 324 and 329, page 18: $\text{C}_2\text{H}_5\text{O}_2\text{COO}^-$.

5. The authors should mention the time or charge with pulsed or constant electrolysis in Fig.2 (b-d).

Response: The corresponding time or charge with pulsed or constant electrolysis has been added in the caption of **Figure 2** in the revised manuscript and was highlighted in red.

Revision:

Line 149-150, page 8: The electrolytic charge of different electrolytic modes is basically kept at about 20 C.

Line 151-152, page 8: Condition optimization of t_{EL} and t_{EH} ($E_{\text{L}} = 0.3 \text{ V}_{\text{RHE}}$, $E_{\text{H}} = 0.7 \text{ V}_{\text{RHE}}$, 50 mM GLY in 1 M KOH) with the selectivity of GLA and the electrolysis time is 1 hour.

6. Page 10, Line 180, “Supplementary Fig. 7 and 7b” should be “Supplementary Fig. 7a and 7b”.

Response: Thanks for the reviewer’s comments. The corresponding revisions have been added to the revised manuscript and were highlighted in red.

Revision:

Line 184, page 10: Supplementary Fig. 7a and 7b.

7. As the author claimed that “the Pt-based catalysts will be deactivated when a high potential ($0.7 V_{\text{RHE}}$) is applied”, why would the further deep oxidation and C–C cleavage still occur? Page 7, Line 122-129.

Response: Thanks for the reviewer’s comments. Catalyst deactivation is a cumulative process; catalyst deactivation does not mean inactivity; if some intermediate species are adsorbed on the surface of the catalyst, it is still possible to be further oxidized, leading to the breakage of the C-C bond. Deactivation of the catalyst leads to fewer active sites, which causes a decrease in the oxidation current (**Figure 2a**).

8. Why did the authors choose graphitic carbon encapsulated Pt nanoparticles as catalysts instead of regular Pt nanoparticles or commercial Pt/C?

Response: In this study, the central role of the graphite shell is to maintain the dispersion of the Pt nanoparticles (preventing nanoparticle aggregation) and the stability of the catalyst (*CCS Chem.* **2022**, 4, 2382–2395; *Nat. Commun.* **2021**, 12, 2002). Meanwhile, the well-contacted graphene layer also facilitates electron transfer during catalysis. The regular Pt nanoparticles or commercial Pt/C undergo aggregation during the catalytic process.

9. In the article, the authors explored the effect of different oxidation potentials (high potential) on glycerol oxidation, but the reduction potentials (low potential) have not been mentioned. In other words, what is the reason the authors would choose $0.3 V_{\text{RHE}}$ as the reduction potential?

Response: Thanks for the reviewer’s comments. In this work, we choose $0.3 V_{\text{RHE}}$ as the reduction potential mainly because the open-circuit potential is close to $0.3 V_{\text{RHE}}$ (**Figure 1d** and **S16**), at which the adsorption state on the catalyst surface can be returned to the state of the open-circuit potential to realize the redistribution of adsorbed species on the catalyst surface. Meanwhile, there is no redox reaction at this potential, which is favorable for us to regulate the catalytic reaction using the pulse potential.

10. As shown in Fig. 2b, at $0.6 V_{\text{RHE}}$ of CE, the selective performance of GLA also can reach 41.7%. What are the main advantages of pulsed potentials?

Response: First of all, the selectivity of GLA at $0.6 V_{\text{RHE}}$ of CE was 41.7%, which is only half of the selectivity of glyceric acid after optimization of the conditions (81%, **Figure 2e**). Secondly, the current retention rate of CE with $0.6 V_{\text{RHE}}$ is 10% after 4000 s of electrolysis, which is much lower than pulse electrolysis (**Figure 2a**, **S5**, and **S10**). Therefore, the main advantages of pulse electrolysis versus constant potential electrolysis are improved product selectivity and mitigated the deactivation of precious metal catalysts during electrolysis.

Reviewer #3:

The authors present a strategy using pulsed potential for the electrocatalytic selective oxidation of glycerol to glyceric acid. In comparison to a constant potential, this approach facilitates the desorption of poisoning intermediates, glyceraldehyde (GLAD), and OH⁻, resulting in higher glyceric acid selectivity. Overall, the chosen research topic is interesting, and the manuscript is logically structured, making it suitable for publication. However, several issues need to be addressed:

i. While both FTIR spectra and COMSOL indicate consistent results, demonstrating a monotonic decrease in the adsorption of aldehyde species on the catalyst surface across experimental conditions PE1 to PE5. However, in experiments, PE3 has the best catalytic performance, not PE5. How to understand this trend and its relationship with electrocatalytic performance?

Response: Thanks for the reviewer's comments. We define PE1-PE5 explicitly in the figure caption of Figure 2. PE1-PE5: $E_H = 0.7 V_{RHE}$ for 0.05 s, $E_L = 0.3 V_{RHE}$ for 2.0, 1.0, 0.5, 0.2 and 0.05 s, respectively. In other words, we keep the oxidation time constant ($t_{EH} = 0.05$ s) and the reduction time (t_{EL}) decreases from 2 s to 0.05 s when the PE protocols range from PE1 to PE5. As we described in the article (Page 17, line 304-308), the adsorption of aldehyde species on the catalyst surface decreased as t_{EL} increased from 0.05 to 2 s (from PE5 to PE1 instead of PE1 to PE5). The same explanation also can be found in the COMSOL analysis section (Page 20, line 343-347). Thus, in terms of GLA selectivity, PE1 ($E_L = 0.3 V_{RHE}$, $E_H = 0.7 V_{RHE}$, $t_{EL} = 2$ s, and $t_{EH} = 0.05$ s) is the most optimal electrolytic condition rather than PE5 ($E_L = 0.3 V_{RHE}$, $E_H = 0.7 V_{RHE}$, $t_{EL} = 0.05$ s, and $t_{EH} = 0.05$ s) (Figure 2d). However, when applied PE1, the duty cycle (t_{EH}/t_{EL}) is only 2.5% under this condition, which results in the lowest GLY oxidation current density (Figure S8-10, Page 10, line 184-194). Therefore, we selected PE3 ($E_L = 0.3 V_{RHE}$, $E_H = 0.7 V_{RHE}$, $t_{EL} = 0.5$ s, and $t_{EH} = 0.05$ s) as the optimal electrolysis condition during our experiments.

ii. According to the description in the Methods section, the reaction energy barriers in Fig. 4d are all simulated using the gas phase molecular model, although the solvation model is added as a correction. However, in the experiment, it is regarded as an electrocatalytic process that occurs on the surface of the catalyst, and the focus is also on the impact of poisoning intermediates on the catalyst surface. Therefore, there is an obvious gap between this simulation and the actual conditions.

Response: We acknowledge the reviewer for pointing out this issue. In the Fig. 4d we focus on calculating the reaction energy barriers for the hydration of aldehydes in an alkaline environment to obtain a geminal diol species (as shown in Formula R1). This reaction is a non-electrochemical process and therefore not only present on the surface of the catalyst, but throughout the electrolyte. Similar analysis also can be found in other related literature (*Angew. Chem. Int. Ed.* **2022**, 61, e202210123; *J. Am. Chem. Soc.* **2020**, 142, 51, 21538–21547).

iii. The atomic structure diagram of "C₂H₅O₂CHO" in Fig. 4e portrays two hydroxyl groups rather than aldehyde groups, contradicting the description as "hexylene glycol" in the text (lines 281-284, page 16). Furthermore, how to understand that the adsorption energy of the dihydroxyl structure is greatly enhanced compared to the single hydroxyl structure (nearly 2 eV stronger as shown in the figure), just because of the addition of one more hydroxyl group?

Response: The literature suggests that, in an alkaline environment, aldehydes exist mainly in geminal diol structure (*Angew. Chem. Int. Ed.* 2022, 61, e202210123; *Energy Environ. Sci.*, 2022, 15, 4175–4189; *J. Am. Chem. Soc.* 2023, 145, 11, 6144–6155). Therefore, we use this geminal diol structure for GLAD adsorption energy calculations in Fig. 4e. The description as "hexylene glycol" in the text is not appropriate and we have changed it to "geminal diol" in the revised manuscript. The structural formula of "C₂H₅O₂CHO" in Fig. 4e has been changed to "C₂H₅O₂CH(OH)₂" (as shown in Figure R4).

Compared to a single hydroxyl structure, the dihydroxyl structure would change the spatial configuration of the entire molecule as well as the adsorbed configuration on the catalyst surface, and this change can lead to a significant increase of the surface adsorption energy.

Figure R4. Adsorption energy of GLY, C₂H₅O₂CH(OH)₂ (transformed from GLAD), and GLA on Pt (111) in an alkaline electrolyte.

Revision:

Line 286, 289 and 291, page 16: **geminal diol**.

iv. The description of the catalysts, "graphitic Pt nanocrystals", lacks clarity (line 61, page 4). It is recommended to specify whether Pt particles are supported on graphitic carbon, capsuled in graphitic carbon, or embedded within the graphitic carbon layer.

Response: In our work, the Pt particles were capsuled in graphitic carbon shell. We have specified it in the revised manuscript and were highlighted in red.

Revision:

Line 60, page 3: In this work, the selective electrocatalytic oxidation of GLY to GLA was achieved through PE, with Pt nanocrystals capsuled in graphitic carbon (Pt@G) as the catalyst.

v. Schematics (Fig. 1e and Fig. 5d) looks a bit confusing. The H atoms are sometimes large and sometimes small; the H of the OH group points the surface, but actually most of the time O points the surface; the functions of high potential and low potential are to promote the oxidation reactions and promote the desorption of poisoning species, respectively, which was not reflected clearly.

Response: We have made improvements to related issues, as shown Fig. 1e and Fig. 5d in the revised manuscript (as shown in Figure R5).

Figure R5. a Schematic of CE-based GEOR over Pt@G. **b** Schematic of the pathways of PE-based GEOR.

Minor issues:

i. In Fig. 2a, several lines are nearly invisible.

Response: In Figure 2a, we mainly compare the stability of the current of CE and PE with different potentials, which is an essential reflection of the stability of the catalyst. In order to show the change in the current retention rate more intuitively and clearly,

we directly extracted the data of the current retention rate after 4000 s of electrolysis for comparison. We made a bar chart in Figure 2a in the revised manuscript (as shown in Figure R6). The corresponding I-t diagram is placed inside the Supporting information (Figure S5).

Figure R6. The current retention rate of CE and PE protocol with different potentials after 4000 s electrolysis.

ii. The seven round white dots in each sub-plot in Fig. 5c lacks an explanation.

Response: The seven round white dots in each sub-plot in Fig. 5c represent the catalyst of Pt@G. Corresponding explanation is given in the caption of the Figure 5c in the revised manuscript and highlighted with red.

Revision:

Line 336-337, page 20: The catalyst of Pt@G is represented by seven round white dots.

iii. In Fig. 2c, the C1 and C2 products should be labeled separately for improved clarity.

Response: We have made improvements to related issues and labeled C1, C2 and C3 separately in Fig. 2 in the revised manuscript (as shown in Figure R7).

Figure R7. Product distribution and conversion rate of GEOR under the electrolytic modes of (a) CE and (b) PE ($E_L = 0.3 V_{RHE}$ for 0.5 s and $E_H = 0.6-1.1 V_{RHE}$ for 0.5 s). The electrolytic charge of different electrolytic modes is basically kept at about 20 C. c Selectivity and GLY conversion rate with 20 mM GLY under the electrolytic mode of PE3.

iv. In Fig. S23, does "DC" represent "constant potential"?

Response: Yes. We have corrected the corresponding mistake of Fig. S23 in the

revised Supporting Information (as shown in Figure R8).

Figure R8. The time-depends surface coverage of different spices. **a** OH^- ; **b** $C_2H_5O_2CH_2OH$; **c** $C_2H_5O_2CHO$; **d** $C_2H_5O_2COOH$.

v. Formulas on Pages 23, 26, and 27 suffer from blurry fonts, making it challenging to assess their correctness.

Response: This could be a problem with the Word version conversion process, which we have fixed.

REVIEWER COMMENTS

Reviewer #1 (Remarks to the Author):

The authors have responded satisfactorily to most of the reviewers' comments. However, there is a need for further clarification on the two points below:

Reviewer 1, comment 1:

The sentence at row 209-212 reading "As shown in Fig 3e, the graphite shells with two layers encapsulated the platinum nanoparticles. Meantime, the graphite shells have defects that allow the platinum nanoparticle core to bind to the substrate and undergo a catalytic reaction" is unclear. I suggest to reformulate it to "The graphite shells encapsulating the platinum nanoparticles have defects and are not complete so that the platinum particles can act as catalysts".

Reviewer 3, comment iv:

It should be clarified that the graphitic layers do not cover the Pt particles completely in the text added to the manuscript. It is Pt that is the catalyst in this study.

Reviewer #3 (Remarks to the Author):

The author has effectively addressed and corrected the majority of the issues raised in this round of responses. However, a concern persists with the second issue. While the hydration of aldehydes into a geminal diol species is a non-electrochemical process, it is crucial to clarify that this does not necessarily imply a predominant occurrence in the electrolyte rather than on the catalyst surface. Notably, the generation of aldehydes from GLY is an electrochemical process and primarily takes place on the catalyst surface. Therefore, assuming that the hydration of aldehydes mainly transpires in the electrolyte demands proof that a significant portion of aldehydes on the surface desorb into the electrolyte, and then, the majority of generated geminal diol species are re-adsorbed to the catalyst surface. Actually, the discussions in this work predominantly revolve around catalyst surface-related phenomena, making the simulation results in the electrolyte hold only quite minor contribution.

Moreover, in the referenced literature for similar analyses indicated by the authors, only the work from the author's research group, *Angew. Chem. Int. Ed.* 2022, 61, e202210123, adopts a calculation model of gas-phase molecules. Contrarily, *J. Am. Chem. Soc.* 2020, 142, 51, 21538–21547, cited from another research group, explicitly states the use of the catalyst surface structure for theoretical modeling. This underscores that the gas-phase molecule model is not a conventional method for studying this. Consequently, modification or removal of this computational model is recommended to prevent potential misinterpretation or confusion.

RESPONSE TO REVIEWERS' COMMENTS

Reviewer #1:

The authors have responded satisfactorily to most of the reviewers' comments. However, there is a need for further clarification on the two points below:

Reviewer 1, comment 1:

The sentence at row 209-212 reading "As shown in Fig 3e, the graphite shells with two layers encapsulated the platinum nanoparticles. Meantime, the graphite shells have defects that allow the platinum nanoparticle core to bind to the substrate and undergo a catalytic reaction" is unclear. I suggest to reformulate it to "The graphite shells encapsulating the platinum nanoparticles have defects and are not complete so that the platinum particles can act as catalysts".

Response: We acknowledge the reviewer's kind suggestion. The corresponding revisions have been added in the revised manuscript and were highlighted in red (Page 11, line 209-211).

Reviewer #3:

The author has effectively addressed and corrected the majority of the issues raised in this round of responses. However, a concern persists with the second issue. While the hydration of aldehydes into a geminal diol species is a non-electrochemical process, it is crucial to clarify that this does not necessarily imply a predominant occurrence in the electrolyte rather than on the catalyst surface. Notably, the generation of aldehydes from GLY is an electrochemical process and primarily takes place on the catalyst surface. Therefore, assuming that the hydration of aldehydes mainly transpires in the electrolyte demands proof that a significant portion of aldehydes on the surface desorb into the electrolyte, and then, the majority of generated geminal diol species are re-adsorbed to the catalyst surface. Actually, the discussions in this work predominantly revolve around catalyst surface-related phenomena, making the simulation results in the electrolyte hold only quite minor contribution.

Comment 2:

Moreover, in the referenced literature for similar analyses indicated by the authors, only the work from the author's research group, *Angew. Chem. Int. Ed.* 2022, 61, e202210123, adopts a calculation model of gas-phase molecules. Contrarily, *J. Am. Chem. Soc.* 2020, 142, 51, 21538 – 21547, cited from another research group, explicitly states the use of the catalyst surface structure for theoretical modeling. This underscores that the gas-phase molecule model is not a conventional method for studying this. Consequently, modification or removal of this computational model is recommended to prevent potential misinterpretation or confusion.

Response: We acknowledge the reviewer's kind suggestion. After careful consideration, we sincerely believe that our original explanation vaguely stated that aldehyde hydration occurred in the bulk electrolyte solution rather than only at the electrode/electrolyte interface is not appropriate. As the reviewer rightly points out, the hydration of the aldehyde group occurs at the electrode/electrolyte interface, owing to the electrochemical reaction primarily takes place on the catalyst surface. In this context, using only gas-phase molecular models to study aldehyde hydration is flawed, even though the hydration reaction is a non-electrochemical process and main influenced by surface alkalinity at the electrode/electrolyte interface. To prevent confusion or misinterpretation, we have removed the gas-phase calculation model for aldehyde hydration in the revised manuscript per the reviewer's suggestion.

Comment 2:

It should be clarified that the graphitic layers do not cover the Pt particles completely in the text added to the manuscript. It is Pt that is the catalyst in this study.

Response: Thanks for the reviewer's comments. The corresponding revisions have been added in the revised manuscript and were highlighted in red.

Revision:

Line 209-211, page 11: The graphite shells encapsulating the platinum nanoparticles have defects and are not complete so that the platinum particles can act as catalysts.

REVIEWERS' COMMENTS

Reviewer #3 (Remarks to the Author):

My concerns have been addressed, and I have no additional comments.